# Clinical Practice Use of Liquid Biopsy to Identify *RAS*/*BRAF* Mutations in Patients with Metastatic Colorectal Cancer (mCRC): A Single Institution Experience

**DOI:** 10.3390/cancers11101504

**Published:** 2019-10-08

**Authors:** Pietro Paolo Vitiello, Vincenzo De Falco, Emilio Francesco Giunta, Davide Ciardiello, Claudia Cardone, Pasquale Vitale, Nicoletta Zanaletti, Carola Borrelli, Luca Poliero, Marinella Terminiello, Gianluca Arrichiello, Vincenza Caputo, Vincenzo Famiglietti, Valentina Mattera Iacono, Francesca Marrone, Alessandra Di Liello, Giulia Martini, Stefania Napolitano, Michele Caraglia, Angela Lombardi, Renato Franco, Ferdinando De Vita, Floriana Morgillo, Teresa Troiani, Fortunato Ciardiello, Erika Martinelli

**Affiliations:** 1Department of Precision Medicine, Università della Campania “Luigi Vanvitelli”, 80131 Napoli, Italy; pietropaolo.vitiello@gmail.com (P.P.V.); fortunato.ciardiello@unicampania.it (F.C.); 2Centro Cellex, Vall D’Hebron Institute of Oncology (VHIO), 08035 Barcelona, Spain; 3Department of Gastrointestinal Medical Oncology, University of Texas, MD Anderson Cancer Center, Houston, TX 77030, USA; 4Department of Experimental Medicine, Università della Campania “Luigi Vanvitelli”, 80138 Napoli, Italy; 5Department of Mental and Physical Health and Preventive Medicine, Pathology Unit, Università della Campania “Luigi Vanvitelli”, 80138 Napoli, Italy

**Keywords:** colorectal cancer, liquid biopsy, *RAS* testing, anti-EGFR, acquired resistance, clonal evolution

## Abstract

Tumor heterogeneity represents a possible cause of error in detecting predictive genetic alterations on tumor tissue and can be overcome by testing alterations in circulating tumor DNA (ctDNA) using liquid biopsy. We assessed 72 consecutive patients with a diagnosis of metastatic colorectal cancer (mCRC) using Idylla™ Biocartis, a fully automated platform that evaluates the most frequent mutations of *KRAS*, *NRAS* and *BRAF* genes. We correlated the results of liquid biopsy and standard tissue-based next generation sequencing (NGS) analyses to patient clinical features. The overall agreement was 81.94%. Concordance was 85.71% and 96.15% in treatment-naïve patients and in the patient subgroup with liver metastases, respectively. In liver metastases positive, treatment-naïve patients, sensitivity, specificity and positive predictive value (PPV) were 92.31%, 100% and 100%, respectively. Circulating mutational fraction (CMF) was significantly higher in patients with liver metastases and high carcinoembryonic antigen (CEA) levels. In a subgroup of patients pre-treated with anti-Epidermal Growth Factor Receptor (EGFR) agents, emerging *KRAS* mutations were evidenced in 33% of cases. Testing *RAS*/*BRAF* mutations on plasma using the Idylla™ Biocartis platform is feasible and reliable in mCRC patients in clinical practice.

## 1. Introduction

Metastatic colorectal cancer (mCRC) is one of the main causes of cancer death worldwide, with 881,000 deaths in 2018 alone [1]. The addition of monoclonal antibodies (MoAbs) to chemotherapy has significantly prolonged survival in mCRC patients compared to chemotherapy alone [2]. Anti-Epidermal Growth Factor Receptor (EGFR) MoAbs, cetuximab and panitumumab, are currently administered in all-*RAS* (*KRAS* and *NRAS*) wild type (WT) mCRC patients as first or subsequent lines of treatment either in monotherapy or in association with chemotherapy, *KRAS* and *NRAS* tumor mutations being negative predictive biomarkers for their use. In fact, the presence of these mutations confers primary resistance to anti-EGFR in mCRC and it is mandatory to test for these mutations on tissue specimen before the initiation of anti-EGFR therapy [3]. Moreover, activating mutations of these genes could also develop during treatment with anti-EGFR in initially WT patients; this phenomenon is known as acquired (or secondary) resistance [4]. Liquid biopsy is an analytical technique consisting of the research of tumor-derived biomarkers in body fluids. Cell-free DNA (cfDNA) in the blood of mCRC patients includes, in different percentages, circulating tumor DNA (ctDNA) released by cancer cells, thus providing potential information in terms of prognosis and prediction of therapeutic sensitivity or resistance. Indeed, analysis of ctDNA has been evaluated in mCRC patients for different purposes: correlation between its levels and survival, monitoring of response to therapy, detection of *RAS*/*BRAF* mutations at different time points [5]. In particular, the theoretical advantage of liquid biopsy over tissue biopsy in the metastatic and/or relapsed disease setting is the possibility to gain a full overview of the genetic make-up of the disease, overcoming both spatial and temporal heterogeneity [6]. In fact, various technologies are now available for testing known mutations in ctDNA in mCRC, the most used are based on digital PCR or quantitative PCR [7]. In particular, digital PCR shows the highest sensitivity, up to a limit of detection of 0.001% for digital droplet PCR [8,9,10] and about 0.1–1% for conventional quantitative PCR [11,12,13]. However, recent reports show how the highest sensitivity of the technique does not reflect a better prediction of the response to anti-EGFR agents, as it may include rare *KRAS* mutant subclones that do not have clinical significance [14,15,16]. The concordance between tissue and liquid biopsies in mCRC, on the other hand, is strictly related not only to the technology used for tissue and plasma analyses, but also to some clinical parameters of the patients, such as the presence or absence of liver metastases [13,17,18,19]. Several retrospective and prospective studies have evaluated the use of liquid biopsy. The main purpose of these studies has generally been the evaluation of the degree of agreement between PCR on formalin-fixed paraffin-embedded (FFPE) tumor tissue and liquid biopsy, especially with beads, emulsion, amplification, magnetics (BEAMing) and digital droplet PCR (ddPCR) techniques [20,21,22,23]. In addition, some research focused on the use of liquid biopsy to guide treatments with anti-EGFR, evaluating the *RAS* status over time and the allelic frequency of mutations of this gene [24,25,26,27]. Similarly, a prospective study by the Franco-British Institute was performed using Idylla™ Biocartis on patients with mCRC, showing an overall agreement with standard-of-care (SoC) of 73%, which increased to 100% in patients with previously untreated metastatic liver disease [28]. Furthermore, in an analysis performed with the Idylla™ Biocartis platform in two first-line prospective clinical trials (PULSE, POSIBA), *RAS* and *BRAF* mutational status was assessed in ctDNA from 178 patients with *KRAS* exon 2 wild-type metastatic colorectal cancer with a sensitivity of 64.1% and a specificity of 90% [29]. Finally, preliminary data from the ongoing ERMES phase III trial in patients with *KRAS*/*NRAS*/*BRAF* WT mCRC also showed concordance of 83.8% of liquid biopsies with standard methods at baseline, with an increased frequency of mutant cases at progression disease (PD) after anti-EGFR treatment [30]. Another study, on the other hand, evaluated the possibility of using anti-EGFR drugs in subsequent lines of therapy, in patients who had *RAS* mutation on the primary tumor and were *RAS* WT on the Idylla™ platform, after treatments with anti-angiogenic drugs [31]. Here we describe our experience with liquid biopsy testing for *KRAS*, *NRAS* and *BRAF* mutations in mCRC, using Idylla™ Biocartis, a fully automated real-time PCR-based platform, in 72 consecutive patients from two groups: patients who had not received prior medical treatment for mCRC (basal cohort) and patients who had received anti-EGFR agents for the treatment of mCRC (post-EGFR cohort).

## 2. Results

### 2.1. Concordance between Tissue and Plasma Samples

A total of 72 patients were analyzed by the Idylla™ Biocartis platform: 47 patients were all-*RAS* and *BRAF* WT, 22 patients were *KRAS* mutated (14 on codon 12, 2 on codon 13, 2 on codon 61, 1 on codon 117, 3 on codon 146), 1 patient was *NRAS* mutated (on codon 61), 2 were *BRAF* mutated (on codon 600). The distribution of clinically relevant characteristics in the study population are shown in Appendix A. The overall concordance between tissue analysis and liquid biopsy was 81.94% (59/72; kappa 0.590; 95% CI: 0.392–0.789). Concordance was also calculated according to different patient subgroups: patients without liver metastases (NON-LIVER) and patients with at least one liver metastasis (LIVER POSITIVE), further divided into patients with metastases limited to the liver (LIVER ONLY) and patients with metastases in the liver and other organs (LIVER and OTHER) (Table 1A). Considering only the liquid biopsies performed on the 42 mCRC patients that were evaluated before starting first line therapy (BASAL COHORT), the concordance raised to 85.71% (36/42; kappa 0.714; 95% CI: 0.507–0.922) with a maximum concordance of 96.15% (25/26; kappa 0.923; 95% CI: 0.776–1) in patients with at least one liver metastasis (LIVER POSITIVE) and 100% in patients with metastases not limited to the liver (LIVER and OTHER) (Table 1B). For more details on the discordant cases in the basal cohort, refer to Table 2.

### 2.2. Sensitivity, Specificity and Positive Predictive Value (PPV)

Overall, sensitivity, specificity and PPV were 77.27%, 84% and 68%, respectively. Analyses of these parameters for different subgroups defined according to patients’ metastatic sites are also shown in Table 1A. The highest sensitivity (100%) was reached in the LIVER and OTHER subgroup with a specificity of 84.62% and a PPV of 71.43%. Sensitivity, specificity and PPV were also calculated in the cohort of treatment-naive patients (BASAL COHORT), considering the subgroups previously described (Table 1B). 

Finally, receiver operating characteristic (ROC) curves for the basal cohort were obtained. The area under the curve (AUC) for the basal patient group was 0.8571 (CI: 0.7335–0.9808), whereas the AUC for the liver metastases positive patient subgroup was 0.9615 (CI: 0.8746–1.000) (Figure 1). 

### 2.3. Circulating Mutational Fraction

The circulating mutational fraction (CMF) was calculated for all the tests that were positive for a *RAS* mutation, representing the relative amount of *RAS* mutated DNA over total cfDNA, as described in the Materials and Methods section. For this reason, we hypothesized that CMF values should be higher for those patients with higher tumor DNA shedding and thus with higher levels of ctDNA over cfDNA. For this purpose, we analyzed the distribution of the CMF values in the 42 mCRC basal cohort patients according to sites of metastases and to CEA level, that were used as a surrogate of tumor burden, as previously described [32]. The median CMF values for the non-liver and for the liver metastases positive subgroup were 0.36% (range 0.0002–10.88%) and 8.24% (range 2.54–23.8%), respectively, with a statistically significant difference between the two subgroups (Figure 2A). In addition, an exploratory analysis evidenced a statistically significant difference between the distribution of the CMF values according to CEA levels above or below the median of the values (46.5 ng/mL) (Figure 2B). 

### 2.4. Post-Anti-EGFR Cohort

This cohort consisted of 30 mCRC patients with *RAS* wild-type tumors on tissue analysis that had received systemic anti-EGFR therapy during treatment for metastatic disease. Liquid biopsy identified 10 *KRAS* mutated patients in this group: 4 codon 61 mutations, 2 codon 146 mutations, 3 codon 12 mutations and 1 case of double positivity for codon 12 and codon 61 mutations. Notably, the distribution of these mutations differs from that found in the 42 mCRC patients that were evaluated before first line treatment, with a higher prevalence of exon 3 and exon 4 as compared to exon 2 mutations, as previously reported [4].

Further, 15 out of these 30 patients were treated with an anti-EGFR drug as a re-challenge treatment strategy. For these patients, serial liquid biopsies were performed during re-challenge therapy. *KRAS* mutations were found during the course of treatment in 5 out of 15 (33.3%) patients. We further assessed the CMF variation of *KRAS* mutations during anti-EGFR re-challenge treatment, comparing the value at the first evidence of *KRAS* mutant circulating clones (T1) and at a following analysis at 6–8 weeks (T2). The continuation of anti-EGFR therapy leads to a substantial and significant increase of CMF in these patients (Figure 3).

## 3. Discussion

The present study has evaluated in a single institution in a clinical practice setting the level of concordance, sensitivity, specificity and PPV of Idylla^TM^ qPCR-based platform by Biocartis on plasma samples in comparison to next generation sequencing (NGS) analysis on tumor specimens in mCRC patients. We have found that these parameters vary according to metastatic sites and according to previous treatments. Concerning the concordance of results, these data are in line with previous reports that have been performed with the Idylla^TM^ Biocartis system [29,30,31], with the concordance higher in mCRC patients for which the analysis has been performed at baseline of metastatic disease (85.71%), especially for those patients with liver metastases (96.15%), who also showed the highest sensitivity (100%). Collectively these data suggest that, in patients with these characteristics, *RAS*/*BRAF* status could effectively be identified by the Idylla^TM^ Biocartis liquid biopsy-based platform. However, the specificity was different between different groups of patients: 84% and 95.24%, respectively in the global and basal cohorts. Collectively, these data suggest that it is possible to identify mutations in the plasma and, therefore, more accurately define the correct mutation status despite a *RAS*/*BRAF* wild-type result on tissue analysis in approximately 1 out of 6 patients and 1 out of 21 patients, respectively in the global and in the basal cohort. Finally, a PPV of 94.12% with the liquid biopsy analysis in the correct detection of mutations in mCRC before first line treatment suggests that this technique is a good alternative to NGS-based testing in clinical practice, especially for those cases in which it is not easy to obtain evaluable tumor tissue. Indeed, the reliability of the test coupled to the very short turn-around time of only about 2 h (85–130 min) from plasma collection to complete mutational report make this technique worthy for this clinical setting. Furthermore, the detection of mutations in the plasma during the course of anti-EGFR-based therapies reflects the emergence and/or the selection of *RAS* mutant clones under the pressure of anti-EGFR MoAbs and, therefore, could potentially allow a dynamic monitoring of disease evolution and efficacy of therapies. For this reason, the analyses for concordance, sensitivity and specificity compared to primary tumor tissue are not useful in the post-EGFR cohort of patients. Notably, in this cohort of patients we found a single case of double positivity (*KRAS* codon 12 and 61). It is known that tumor heterogeneity could lead to the development of several mutated clones, but the clinical relevance of them is still debated. In the work by Thierry et al. [33], a higher percentage of samples harbored more than one mutation, but this reflects the higher sensitivity of IntPlexVR used for ctDNA analysis; in particular, the lower the threshold, the higher the percentage of more-than-one mutation detected. Idylla^TM^ has a lower sensitivity but this could probably eliminate a “background noise” of clinically irrelevant mutations. 

Finally, to our knowledge, this study provides the first evidence that by using the Idylla^TM^ Biocartis platform for circulating tumor DNA it is possible to calculate the CMF for those cases positive for *RAS* or *BRAF* mutation. This value represents the fraction of mutant DNA over total cfDNA and, as such, it highly depends on the quantity of non-tumor circulating DNA. Nonetheless, CMF is directly correlated with both the abundance of total ctDNA and the mutation allele fraction (MAF). Although further prospective data are needed in this context, CMF might be used as a practical clinical surrogate of these two parameters. 

In the 42 basal mCRC patient cohort, this value significantly correlates with the presence of liver metastases and with higher CEA levels, the latter being used as a surrogate for tumor burden. Further, even with the limitation of a small sample size (a total of five patients), the CMF of acquired *KRAS* mutations increases over time in patients treated with anti-EGFR agents as re-challenge therapy. In the frame of a serial testing for *RAS* mutations in liquid biopsies in patients undergoing anti-EGFR therapy, a raise in CMF levels might represent a potential marker of cancer cell resistance onset and might anticipate the clinical evidence of disease progression and, thus, might allow a rational change in cancer therapy.

### Learning from the Discordant Cases

Among the discordant cases in the 42 mCRC patient basal cohort (Table 2), we have identified three cases (patients A–C) with a *KRAS* mutation in NGS tissue analysis and *RAS* WT status in liquid biopsy. These three patients did not have liver metastases, a condition that is more often associated with less ctDNA shedding in the blood. In fact, as confirmed in other works using even more sensitive techniques, the absence of liver metastases was the also main clinical factor associated with inconclusive circulating tumor DNA results in previous reports [19]. 

A fourth case of *RAS* mutation identified in NGS tissue analysis but not in liquid biopsy analysis (patient D) refers to a patient with low tumor burden (multiple subcentimetric liver metastases, not amenable to resection). Furthermore, since this was a *NRAS* mutation, the sensitivity of the Idylla^TM^ Biocartis platform to identify *NRAS* mutations is not high enough for such a case, since it requires at least 5% of allelic frequency for optimal detection of these mutations [34]. Interestingly, when a subsequent plasma analysis was performed at disease progression, the same *NRAS* mutation previously evidenced in NGS tissue testing at baseline (Q61H) was detected, suggesting cancer cell clonal expansion. A *BRAF* V600E mutation was detected in NGS tissue analysis but not in liquid biopsy in a case (patient E) with low tumor burden and lung limited metastatic disease. Similarly to *NRAS* mutations, the optimal detection for *BRAF* V600 mutations using Idylla^TM^ Biocartis platform for plasma is above 5% of mutated allele frequency [34]. Interestingly, in one case (patient F), *NRAS* mutation was detected in liquid biopsy at diagnosis of tumor recurrence, while the NGS analysis of primary tumor tissue resulted in *RAS* wild type. This patient was treated with anti-EGFR agents in first line as the standard of care but experienced a disease progression at the first evaluation after only 3 months of therapy, as generally occurs with primary resistance. This case underlines the potential advantage of performing liquid biopsy at the time of tumor recurrence as compared to tissue analysis of the primary tumor for a better clinical prediction on the efficacy of anti-EGFR therapies. 

## 4. Materials and Methods 

### 4.1. Study Oversight

Patients provided informed consent for an institutional review board-approved protocol for longitudinal collection of plasma and profiling on tumor DNA within the I-Cure research program. Between 1 July 2018 and 30 June 2019, 72 patients were evaluable for analysis. Among them, 42 patients were assessed at baseline (i.e., before the initiation of any systemic treatment for mCRC): basal cohort; whereas 30 patients with all *KRAS*/*NRAS* WT on tissue analysis were treated with a therapy regimen including an anti-EGFR MoAb (cetuximab or panitumumab) and analyzed for liquid biopsy during the course of treatment for metastatic disease: post-EGFR cohort. 

### 4.2. Plasma Collection

At least 6 mL of whole blood was collected by standard procedure peripheral vein blood draw, using Vacutainer^®^ with EDTA as anticoagulant (K2EDTA, purple cap, catalog #367863, Becton Dickinson). Plasma was separated through two different centrifugation steps (the first at room temperature for 10 min at 1500× *g* and the second at 2000× *g* for the same time and temperature). Plasma was stored at −80 °C until analysis.

### 4.3. Mutational Analyses of Tissue Specimens

Analyses of tissue specimens were all performed in the Pathology Service of Università della Campania “Luigi Vanvitelli” using Next Generation Sequencing (NGS). 

#### 4.3.1. Sample Preparation

An appropriate formalin-fixed paraffin-embedded (FFPE) tissue block was selected for each case. Four unstained FFPE tissue sections were cut at 10 μm each for DNA extraction. DNA was obtained using the QIAamp^®^DNA FFPE kit Tissue (Qiagen, Duesseldorf, Germany), according to the manufacturer’s instructions. Extracted DNA was eluted in 20 or 30 µL of elution buffer and then DNA was quantified by a Qubit^®^ 2.0 Fluorometer (Life Technologies, Singapore) using the Qubit^®^ dsDNA HS Assay kit, according to the manufacturer’s recommendations. The extracted DNA was stored at −20 °C. The Ion Torrent Personal Genome Machine (PGM) technology allows the massive parallel sequencing of DNA libraries, of several different samples, using an approach based on the PH variations that occur at the moment of incorporation of the single deoxyribonucleotide into the reaction catalyzed by the DNA polymerase.

#### 4.3.2. Library Preparation

Around 10 ng of DNA was used to prepare the sequencing libraries. The libraries were prepared with the IonAmpliSeq™ Library kit 2.0 (Thermo Fisher Scientific, Carlsbad, CA, USA) and with primer pool: IonAmpliSeq Colon and Lung Cancer Research Panel v2, which analyzes 504 mutational hotspots and targeted regions in 22 genes, including *KRAS*, *BRAF* and *NRAS*.

Amplified products were purified with Agencourt AMPure XP beads (Beckman Coulter Genomics, High Wycombe, UK). Concentrations of amplified and bar-coded libraries were measured using the Qubit ^®^ 2.0 Fluorometer (Life Technologies, Singapore) and the Qubit^®^ dsDNA HS Assay kit (Life Technologies, Singapore). DNA libraries were stored at −20 °C. The libraries were clonally amplified on Ion Sphere™ particles after dilution of the libraries to 100 pM. Template preparation was performed with the IonOneTouch™ 2 System (Thermo Fisher Scientific, Carlsbad, CA, USA), an automated system for emulsion PCR, recovery of Ion Sphere™ particles and enrichment of template-positive particles. The Ion Sphere™ particles coated with template were applied to the semiconductor chip. A short centrifugation step was conducted to allow the spherical particles to be deposited into the chip wells. Finally, sequencing was carried out using Ion 316™ chips on the Ion Personal Genome Machine System (PGM™, Thermo Fisher Scientific) using the Ion PGM™ Hi-Q view Sequencing kit v2 (Thermo Fisher Scientific, Carlsbad, CA, USA). 

#### 4.3.3. Data Analysis

The Torrent Suite Software v.4.0.2 (Life Technologies) was used to assess run performance and data analysis. Integrative Genomics Viewer (IGV v 2.2, Broad Institute) was used for visual inspection of the aligned reads. Sequencing data were analyzed using Ion Reporter software (https://ionreporter.lifetechnologies.com/) and further filtered through quality checking. We selected all single nucleotide variants (SNVs) in the studied genes resulting in a non-synonymous amino acid change, or a premature stop codon, and all short indels resulting in either a frameshift or insertion/ deletion of amino acids. All SNVs were analyzed for previously reported hotspot mutations (somatic mutations reported in COSMIC database(cancer.sanger.ac.uk) and novel variations, i.e., new mutations detected by NGS but not reported in either COSMIC or db SNP databases. 

### 4.4. Mutational Analyses of Plasma

Analyses of plasma were all carried out using the automated Idylla™ qPCR-based platform by Biocartis. Briefly, 1 mL of fresh plasma was added to the specific cartridge for automated analysis. The whole procedure takes less than 1 min hands-on time and about 2 h for the analysis (85 to 130 min). ct*KRAS* and ct*NRAS*/*BRAF*/EGFR cartridges were used. The mutations detected by the cartridges are: ct*KRAS*: exon 2 (G12 > C/R/S/A/D/V, G13 > D), exon 3 (A59 > E/G/T, Q61 > K/L/R/H), exon 4 (K117 > N, A146 > P/T/V); ct*NRAS*/*BRAF*/EGFR: *NRAS* exon 2 (G12 > C/S/A/D/V, G13 > D/V/R), exon 3 (A59 > T, Q61 > K/L/R/H), exon 4 (K117 > N, A146 > T/V), *BRAF* V600 > E/D/K/R, EGFR S492 > R. The limit of detection of the Idylla^TM^ platform for cfDNA is set at about 10,000 copies of WT DNA (about 30 ng) per milliliter. At the end of the run the platform displays an automated report that includes: validity of the run (in case no DNA is detected, the test is considered invalid), presence or absence of mutation(s), type of mutation(s) detected, Cq of the total *KRAS*/*NRAS*/*BRAF*, Cq of the specific mutation(s) detected. No specific training is required in order to interpret the test result. 

In order to evaluate the reproducibility of this test in our series, we collected plasma samples of the first 10 patients in duplicate at a short time interval (6–48 h). Mutation analysis with Idylla^TM^ platform was performed twice for this set of samples: if the result had been discordant, the analysis would have been repeated for the third time. Considering 100% concordance between two consecutive samples in the first set of 10 patients, no additional analysis was performed for the following cases. Additionally, in the case of discordance between plasma and tissue-based or previous plasma-based analyses, a confirmatory plasma sample was acquired and re-analyzed.

### 4.5. Concordance

Accuracy was calculated using the interrate agreement kappa (K) coefficient [35]. The strength of agreement was interpreted according to K value as follows: poor (<0.20), fair (0.21–0.40), moderate (0.41–0.60), good (0.61–0.80) and very good (0.81–1.00). All analyses were performed using Graphpad online Kappa calculator.

### 4.6. Sensitivity, Specificity and Positive Predictive Value

Assuming the NGS analysis of FFPE as gold standard analysis, the results of the liquid biopsy analysis were defined as true positive (TP) when the same mutation was found in both NGS and liquid biopsy analyses, true negative (TN) when no mutation was detected by both analyses, false positive (FP) when a specific mutation was only found in liquid biopsy analysis and false negative (FN) when a specific mutation was detected only by NGS of FFPE. Sensitivity was defined as the ability of the liquid biopsy platform to detect mutations in patients with known alterations in *RAS*/*BRAF* genes by NGS analysis of FFPE (sensitivity = TP/(TP + FN)). Specificity was defined as the proportion of patients without mutations in both tissue and plasma among patients without known genetic alterations by NGS analysis of FFPE (specificity = TN/(TN + FP)). Positive predictive value (PPV) was defined as the probability that a mutation detected in the plasma sample was also present in the tissue sample (PPV = TP/(TP + FP)).

### 4.7. Circulating Mutational Fraction

ΔCq values, defined as the difference between the Cq of the mutation and the mean Cq of the internal control, were obtained for each test that was positive for a mutation. The percentage of mutated DNA over the internal control was calculated as follows: 2 − ΔCq × 100; this value was defined as the circulating mutational fraction (CMF). In the basal cohort, median CMF values were compared between the liver metastases positive patient group and the liver metastases negative group, using the Mann–Whitney U test. Median CMF values were also compared according to CEA levels, by using the same test. In the subgroup of patients undergoing anti-EGFR re-challenge therapy, the CMF values of the acquired *RAS* mutations were compared between the first evidence and the re-analysis at 6–8 weeks, by using the Wilcoxon rank sum test. All statistical analyses were performed using Graphpad (Prism 8).

## 5. Conclusions

We have shown how using a simple PCR-based assay for *RAS* and *RAF* testing in plasma of mCRC patients is convenient and clinically relevant. Taking into account the reliability and the repeatability of the described method, as well as the short turn-around time and the sustainable costs [36], this approach could be feasible and effective in the clinical practice management of mCRC patients.

## Figures and Tables

**Figure 1 cancers-11-01504-f001:**
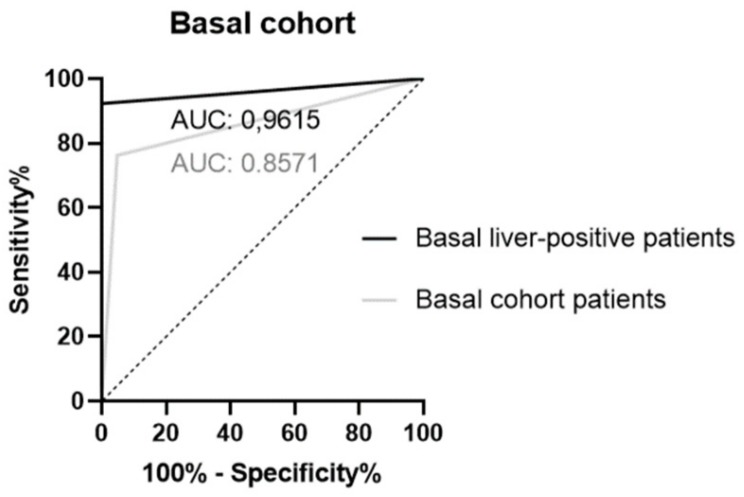
Receiver operating characteristic (ROC) curves for the BASAL COHORT patients (light grey, AUC: 0.8571, CI: 0.7335–0.9808) and for the subgroup of basal LIVER POSITIVE patients (black, AUC: 0.9615, CI: 0.8746–1.000).

**Figure 2 cancers-11-01504-f002:**
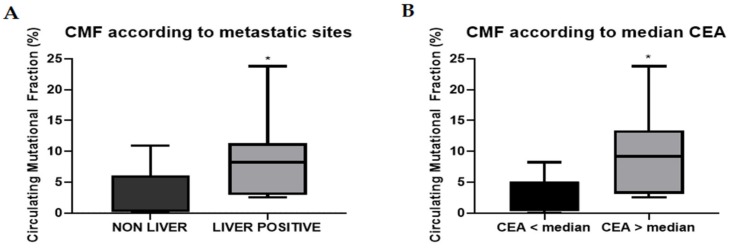
Circulating mutational fraction (CMF) distribution between clinical subgroups. (**A**) Difference between NON-LIVER (5 pts) and LIVER POSITIVE (9 pts) metastatic colorectal cancer (mCRC) (*p* = 0.0385). (**B**) Difference between patients with baseline CEA levels below (7 pts) or above (7 pts) median (*p* = 0.0469). *: *p* < 0.05. NON-LIVER: patients with no evidence of liver metastases. LIVER POSITIVE: all the patients with at least one liver metastasis (the sum of LIVER ONLY and LIVER and OTHER).

**Figure 3 cancers-11-01504-f003:**
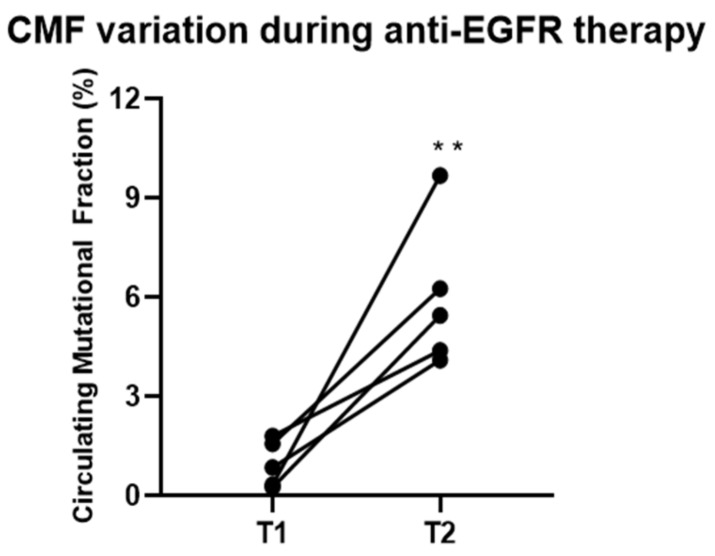
CMF variation during anti-Epidermal Growth Factor Receptor (EGFR) re-challenge treatment (*p* = 0.0079). T1: first evidence of *KRAS* mutant circulating clones, T2: analysis of *KRAS* circulating mutations 6–8 weeks after T1. **: *p* < 0.01.

**Table 1 cancers-11-01504-t001:** Concordance, sensitivity, specificity and positive predictive value according to patients’ cohort. (A) GLOBAL COHORT; (B) BASAL COHORT. NON-LIVER: patients with no evidence of liver metastases; LIVER ONLY: patients with metastases limited to the liver; LIVER and OTHER: patients with metastases not limited to the liver; LIVER POSITIVE: all the patients with liver metastases (the sum of LIVER ONLY and LIVER and OTHER); TP: true positive; TN: true negative; FP: false positive; FN: false negative; PPV: positive predictive value.

**A**	**N°**	**TP**	**TN**	**FP**	**FN**	**Concordance %**	**Sensitivity %**	**Specificity %**	**PPV %**
**Global cohort**	72	17	42	8	5	81.94	77.27	84	68
Non-liver	23	4	13	2	4	73.91	50	86.67	66.67
Liver positive	49	12	30	6	1	85.71	92.31	83.33	66.67
Liver only	13	2	8	2	1	76.92	66.67	80	50
Liver and other	36	10	22	4	0	88.89	100	84.62	71.43
**B**	**N°**	**TP**	**TN**	**FP**	**FN**	**Concordance %**	**Sensitivity %**	**Specificity %**	**PPV %**
**Basal cohort**	42	16	20	1	5	85.71	76.19	95.24	94.12
Non-liver	16	4	7	1	4	68.75	50	87.5	80
Liver positive	26	12	13	0	1	96.15	92.31	100	100
Basal liver only	8	3	4	0	1	87.5	75	100	100
Basal liver and other	18	9	9	0	0	100	100	100	100

**Table 2 cancers-11-01504-t002:** Discordant cases in the basal cohort (see text for details). MAF: mutation allele fraction; WT: wild type.

Case	Results of Tissue	Results of Liquid Biopsy	Clinical Characteristics	Possible Explanations
A	*KRAS* MUT (G12C)	*KRAS* WT, *NRAS* WT, *BRAF* WT	Pelvic-infiltrating, inoperable rectal cancer. No distant metastasis.	The absence of distant metastases is associated with low abundance of circulating tumor DNA (ctDNA).
B	*KRAS* MUT (G13D)	*KRAS* WT, *NRAS* WT, *BRAF* WT	Left colon cancer with loco-regional disease.	The absence of distant metastases is associated with low abundance of circulating tumor DNA (ctDNA).
C	*KRAS* MUT (G12V)	*KRAS* WT, *NRAS* WT, *BRAF* WT	Left colon cancer with subcentimetric nodal disease.	The absence of distant metastases is associated with low abundance of circulating tumor DNA (ctDNA).
D	*NRAS* MUT (Q61H)	*KRAS* WT, *NRAS* WT, *BRAF* WT	Left colon cancer with multiple subcentimetric liver metastases.	Low MAF for this mutation on ctDNA (predicted sensitivity is optimal for MAF > 5%). The liquid biopsy performed at disease progression (tumor burden increased) confirmed *NRAS* mutation.
E	*BRAF* MUT (V600E)	*KRAS* WT, *NRAS* WT, *BRAF* WT	Right colon cancer with multiple centimetric lung metastases.	Patient with non-liver metastatic disease and a low burden of disease. Low MAF for this mutation on ctDNA (predicted sensitivity is optimal for MAF > 5%).
F	*KRAS* WT, *NRAS* WT, *BRAF* WT	*NRAS* MUT (G13D)	Nodal-limited recurrence of left colon cancer.	The relapsed cancer is enriched with *NRAS* mutant cells, that were missed on tissue analysis.

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
