# Peer review of "Clinical Practice Use of Liquid Biopsy to Identify RAS/BRAF Mutations in Patients with Metastatic Colorectal Cancer (mCRC): A Single Institution Experience"

_cancers, 2019, doi:10.3390/cancers11101504_

Round 1

Reviewer 1 Report

As the authors point out, colorectal cancer (CRC) is one of the leading causes of cancer deaths worldwide.

Anti-EGFR mAb (cetuximab or panitumumab) and anti-VEGF mAb (bevacizumab) are the two main targeted agents available for RAS wild-type (WT) metastatic colorectal cancer (mCRC) treatment.

Nonetheless, three head-to-head clinical trials evaluating anti-EGFR mAb vs -VEGF mAb in first-line treatment failed to conclude a uniform result. Recently, a few small clinical studies revealed that prior use of bevacizumab might impair the effect of cetuximab or panitumumab. Preclinical studies have also suggested that pretreatment with bevacizumab may lead to simultaneous resistance to anti-EGFR mAb. 

In this context the heterogeneity of the tumor represents a serious obstacle to the understanding of the important genetic alterations and therefore to the prediction of the best therapy. Furthermore, acquired or constitutive resistance is a major hindrance to cytotoxic and molecular anticancer therapies.

Therefore the manuscript treats an interesting topic and the experimental techniques used are at the forefront of technology. Furthermore the paper is well written and the tables and figures are clear. Statistical analyses were performed correctly.

In summary, the authors show that the RAS/BRAF genetic mutations assay on liquid biopsy in patients with metastatic colorectal cancer (mCRC) could be useful to predict the patient’s response to specific treatments and to early identify possible development of resistance.

Overall, I think this paper has reinforced the idea that the analysis of RAS and RAF mutations in plasma of mCRC patients is suitable and clinically relevant.

General comments

The issue of resistance is very important for the proper management of patients.

The datum from the patient called F (Table2) in which the NRAS mutation was identified on the liquid biopsy while it was missing on the tissue analysis is very impressive. I agree with the authors that the development of resistance in this patient after only 3 months of anti-EGFR therapy, as is the case with primary resistance, is a strong demonstration of how the liquid biopsy can soon be very instructive about patient response to treatment. A greater number of patients with these characteristics would have allowed us to better confirm this feeling.

Also noteworthy is the case of the patient called D (Table 2). He showed the NRAS mutation on tissue biopsy while he did not show it on the liquid biopsy. Conversely the liquid biopsy performed at disease progression confirmed it. This may be interesting to suggest the prognosis and allow a change of therapy.

Overall I believe that the 94.12% as PPV in the detection of mutations on liquid biopsy in mCRC patients before first-line treatment is strongly indicative that this technique could be alternative, or at least supportive, to FFPE tumor tissue analysis. However I do not agree with the authors on the methodological choice to compare the study of mutations on FFPE tissues and on liquid biopsies with different methods, NGS (for FFPE tissues) vs Idylla platform (for liquid biopsy). The Idylla platform allows the study also on FFPE tissue samples; a comparative analysis using the same technology would have been preferable. The authors should explain this methodological choice.

Specific comments:

In the text at line 111: the number of patients with metastasis both in the liver and in other organs is 36 (replace 23 with 36).

In figure 3: referring to “KRAS mutations were found during the course of treatment in 5 out of 15 patients”. It appears that the authors reported data on changes in CMF during anti-EGFR therapy for only 4 patients. They should also add data from the 5th patient.

-------------------------------------------------------------------------------------

Author Response

Dear Reviewer, 

thank you for considering our work for publication. 

Here are listed the responses to the comments as a point by point list, with the Authors' responses marked in Bold Italics

As the authors point out, colorectal cancer (CRC) is one of the leading causes of cancer deaths worldwide.

Anti-EGFR mAb (cetuximab or panitumumab) and anti-VEGF mAb (bevacizumab) are the two main targeted agents available for RAS wild-type (WT) metastatic colorectal cancer (mCRC) treatment.

Nonetheless, three head-to-head clinical trials evaluating anti-EGFR mAb vs -VEGF mAb in first-line treatment failed to conclude a uniform result. Recently, a few small clinical studies revealed that prior use of bevacizumab might impair the effect of cetuximab or panitumumab. Preclinical studies have also suggested that pretreatment with bevacizumab may lead to simultaneous resistance to anti-EGFR mAb. 

In this context the heterogeneity of the tumor represents a serious obstacle to the understanding of the important genetic alterations and therefore to the prediction of the best therapy. Furthermore, acquired or constitutive resistance is a major hindrance to cytotoxic and molecular anticancer therapies.

Therefore the manuscript treats an interesting topic and the experimental techniques used are at the forefront of technology. Furthermore the paper is well written and the tables and figures are clear. Statistical analyses were performed correctly.

In summary, the authors show that the RAS/BRAF genetic mutations assay on liquid biopsy in patients with metastatic colorectal cancer (mCRC) could be useful to predict the patient’s response to specific treatments and to early identify possible development of resistance.

Overall, I think this paper has reinforced the idea that the analysis of RAS and RAF mutations in plasma of mCRC patients is suitable and clinically relevant.

General comments

The issue of resistance is very important for the proper management of patients.

The datum from the patient called F (Table2) in which the NRAS mutation was identified on the liquid biopsy while it was missing on the tissue analysis is very impressive. I agree with the authors that the development of resistance in this patient after only 3 months of anti-EGFR therapy, as is the case with primary resistance, is a strong demonstration of how the liquid biopsy can soon be very instructive about patient response to treatment. A greater number of patients with these characteristics would have allowed us to better confirm this feeling.

Also noteworthy is the case of the patient called D (Table 2). He showed the NRAS mutation on tissue biopsy while he did not show it on the liquid biopsy. Conversely the liquid biopsy performed at disease progression confirmed it. This may be interesting to suggest the prognosis and allow a change of therapy.

Overall I believe that the 94.12% as PPV in the detection of mutations on liquid biopsy in mCRC patients before first-line treatment is strongly indicative that this technique could be alternative, or at least supportive, to FFPE tumor tissue analysis. However I do not agree with the authors on the methodological choice to compare the study of mutations on FFPE tissues and on liquid biopsies with different methods, NGS (for FFPE tissues) vs Idylla platform (for liquid biopsy). The Idylla platform allows the study also on FFPE tissue samples; a comparative analysis using the same technology would have been preferable. The authors should explain this methodological choice.

We thank the Reviewer for the precious comments and suggestions. The choice of different methodologies for detecting mutations on tissue FFPE specimen and plasma stems from the intent to compare a very sensitive methodology in tissue analysis, such as NGS  that is now widespread in our routine clinical practice, to a very easy-to-perform, albeit less sensitive, methodology as Idylla. In fact, both tissue and plasma analyses by Idylla are PCR-based. The comparison between tissue and plasma qPCR-based analyses  goes beyond the scope of our work.

Specific comments:

In the text at line 111: the number of patients with metastasis both in the liver and in other organs is 36 (replace 23 with 36).
We corrected the text accordingly.

In figure 3: referring to “KRAS mutations were found during the course of treatment in 5 out of 15 patients”. It appears that the authors reported data on changes in CMF during anti-EGFR therapy for only 4 patients. They should also add data from the 5th patient.
As the Reviewer rightly suggests, a previous version of figure 3 was mistakenly used for submission. An updated version of the figure is now present, including all 5 patients.

Reviewer 2 Report

In their manuscript titled “Clinical practice use of liquid biopsy to identify RAS/BRAF mutations in patients with metastatic colorectal cancer (mCRC): a single Institution experience”, the author studied the use of the qPCR-based platform “Idylla Biocartis” to identify frequent mutations found in KRAS, NRAS and BRAF genes in plasma samples in a clinical setting. Results obtained from liquid biopsy and standard tissue-based NGS were correlated to clinical features, e.g. presence and location of metastases. Out of the 72 patients with diagnosed mCRC, the mutational RAS/BRAF status was determined in 42 patients before start of therapy. 30 mCRC patients had received a prior treatment with anti-EGFR antibodies before RAS/BRAF status was analyzed. The concordance between the mutational status of primary tumor and plasma samples was 81.9% for the total (“gobal”) cohort, whereas the concordance was 73.9% in patients without liver metastases, 76.9% in patients with liver metastases only, and 88,9% in patients with metastases in liver and other organs, respectively. Out of the 42 mCRC patients that were untreated before start of therapy and showed liver metastases (n=26), sensitivity, specificity and positive predictive value were 100%, 92,9%, and 92,3%, respectively. Concordance was 96,1% for the mutational status between primary tumor and plasma samples of these patients. Based on the relative amount of RAS mutated DNA over total cfDNA, a “circulating mutational fraction” (CMF) was defined. The CMF was significantly higher in patients with liver metastases. Furthermore, the authors suggest that an increase in the CMF levels might indicate a beginning cancer cell resistance and, hence, be a potential marker. Based on their findings, the authors conclude that analyzing the RAS/BRAF mutations in plasma samples using the “Idylla Biocartis” platform is feasible in a clinical setting and delivers reliable data. The presented data is interesting. However, the study is somewhat limited by its focus on the “Idylla Biocartis” qPCR platform. It would be interesting to find out how this system performs in comparison to other qPCR-based or digital droplet PCR (ddPCR)-based systems. Given its use in a clinical setting it would also be of interest how reproducible the detection of mutations is in a given set of samples in repeated measurements. In summary, I recommend the publication of this manuscript after the major and minor points have been addressed (see below for details). Major points: 1) Please the describe the “standard procedure” for collecting blood in more detail. a. What kind of tubes were used (company? product? catalog number?) b. What was the average / maximum volume of blood drawn? c. How much plasma / cfDNA did you yield from 6 mL of blood? 2) What is the yield of cfDNA of each patient plasma sample? What is the minimal amount of cfDNA that can be used to perform the analysis on the “Idylla Biocartis” system? Please provide a (supplementary) table listing the amount of cfDNA for each patient sample. 3) Please provide specific information how the NGS was performed. a. Which NGS platform / machine was used? b. What reagents were used? c. How was the analysis of sequence data performed? d. What was the coverage? e. Which software was used for the analysis? 4) The concordance between NGS (primary tumor) and qPCR data (plasma sample) is one important aspect to judge the quality of the results obtained by using the “Idylla Biocartis” platform. Another important criterion is the reproducibility of the qPCR-based analysis in a given set of plasma samples. Do repeated measurements of the RAS/BRAF mutational status yield the same result in a given sample? Please present the corresponding data in your manuscript. 5) In the introduction, the authors state that “using the Idylla Biocartis platform … is feasible in clinical practice.” One criteria to support this statement is the time for performing the analysis and interpret the data. How long does it take to prepare the samples, run the qPCR, analyze the data? How easy/difficult is it to interpret the data? How long does it take to train a person to perform the testing of RAS/BRAF mutational status? Please add the requested information to the discussion of the manuscript. 6) Figure 2A: Sensitivity, specificity, and ppv can be used to describe the quality of a given marker. However, it is difficult to “classify” a single patient based on the analysis on the RAS/BRAF status with respect to his/her prognosis or response to therapy. In this context, the CMF value could be helpful. Does the “basal liver positive” (Table 1B) cohort contain only patients with liver metastasis only or does it contain patients with lover metastases only and patient with metastases in liver and other organs? a. Please define the “basal liver positive” cohort in Table 1B and add this information to the table legend. b. Please define the “basal liver positive” cohort in Figure 2A and add this information to the figure legend. c. Please use the same labels in Table 1B and Figure 2A and add the number of patients in each group to the figure legend of Figure 2A. d. If the “basal liver positive” cohort contains patients with lover metastases only and patient with metastases in liver and other organs, please change Figure 2A and distinguish between “liver positive” and “liver & other” by adding a third box to the diagram. Minor points: 1) Line 133: Please change “Figure 2 B” to “Figure 2B”.

Author Response

Dear Reviewer, 

thank you for considering our work for publication. 

Here are listed the responses to the comments as a point by point list, with the Authors' responses marked in Bold Italics

In their manuscript titled “Clinical practice use of liquid biopsy to identify RAS/BRAF mutations in patients with metastatic colorectal cancer (mCRC): a single Institution experience”, the author studied the use of the qPCR-based platform “Idylla Biocartis” to identify frequent mutations found in KRAS, NRAS and BRAF genes in plasma samples in a clinical setting. Results obtained from liquid biopsy and standard tissue-based NGS were correlated to clinical features, e.g. presence and location of metastases. Out of the 72 patients with diagnosed mCRC, the mutational RAS/BRAF status was determined in 42 patients before start of therapy. 30 mCRC patients had received a prior treatment with anti-EGFR antibodies before RAS/BRAF status was analyzed. The concordance between the mutational status of primary tumor and plasma samples was 81.9% for the total (“gobal”) cohort, whereas the concordance was 73.9% in patients without liver metastases, 76.9% in patients with liver metastases only, and 88,9% in patients with metastases in liver and other organs, respectively. Out of the 42 mCRC patients that were untreated before start of therapy and showed liver metastases (n=26), sensitivity, specificity and positive predictive value were 100%, 92,9%, and 92,3%, respectively. Concordance was 96,1% for the mutational status between primary tumor and plasma samples of these patients. Based on the relative amount of RAS mutated DNA over total cfDNA, a “circulating mutational fraction” (CMF) was defined. The CMF was significantly higher in patients with liver metastases. Furthermore, the authors suggest that an increase in the CMF levels might indicate a beginning cancer cell resistance and, hence, be a potential marker. Based on their findings, the authors conclude that analyzing the RAS/BRAF mutations in plasma samples using the “Idylla Biocartis” platform is feasible in a clinical setting and delivers reliable data. The presented data is interesting. However, the study is somewhat limited by its focus on the “Idylla Biocartis” qPCR platform. It would be interesting to find out how this system performs in comparison to other qPCR-based or digital droplet PCR (ddPCR)-based systems. Given its use in a clinical setting it would also be of interest how reproducible the detection of mutations is in a given set of samples in repeated measurements. In summary, I recommend the publication of this manuscript after the major and minor points have been addressed (see below for details).

The Authors thank the Reviewer for the interesting comments on the manuscript.

We agree with the Reviewer’s interest in comparing different methodologies for detecting circulating tumour DNA and performing analyses (i.e. qPCR, ddPCR). However, since the aim of the work is to prove the clinical relevance and feasibility of liquid biopsy in guiding targeted therapy in metastatic colorectal cancer patients, making a comparison between different techniques goes beyond the scope of the paper. Moreover, one study that figures in the reference list (Vivancos A. et al, Sci Rep 2019) [ref.13] has already provided data on the concordance between Idylla Biocartis (qPCR-based) and OncoBEAM (dPCR-based) plasma analysis. In addition to that, since Idylla biocartis is a fully automated platform some major molecular information are not available to the Authors, including ctDNA dosing.

 Major points:

1) Please the describe the “standard procedure” for collecting blood in more detail. a. What kind of tubes were used (company? product? catalog number?) b. What was the average / maximum volume of blood drawn? c. How much plasma / cfDNA did you yield from 6 mL of blood?

This information has been added to the original manuscript and is now available. In particular:

6 ml K-EDTA tubes (Becton Dickinson, Vacutainer K2EDTA purple cap, Catalog #367863) were used; The average volume of blood drawn was 6 ml, with a maximum of 9 ml; The mean yield of plasma was about 3 ml for each patient. cfDNA yield was not calculated, as the automated platform does not provide this information.

2) What is the yield of cfDNA of each patient plasma sample? What is the minimal amount of cfDNA that can be used to perform the analysis on the “Idylla Biocartis” system? Please provide a (supplementary) table listing the amount of cfDNA for each patient sample.

As stated before, we are not able to provide this information. In fact, a biological study that aims to validate the amount of cfDNA from each patient goes beyond the scope of our work and would need supplementary plasma to be performed. However, the instruction for use manual of the Idylla ctKRAS cartridges report a limit of detection (LOD) of about 30 ng (about 10.000 copies) of WT DNA [ref. IFU ctKRAS IVD 1.0].

This information was added to the Materials and Methods section.

3) Please provide specific information how the NGS was performed. a. Which NGS platform / machine was used? b. What reagents were used? c. How was the analysis of sequence data performed? d. What was the coverage? e. Which software was used for the analysis?

The required information has been added to the manuscript in the Materials and Methods section.

4) The concordance between NGS (primary tumor) and qPCR data (plasma sample) is one important aspect to judge the quality of the results obtained by using the “Idylla Biocartis” platform. Another important criterion is the reproducibility of the qPCR-based analysis in a given set of plasma samples. Do repeated measurements of the RAS/BRAF mutational status yield the same result in a given sample? Please present the corresponding data in your manuscript.

We thank the reviewer for this very important observation.

It is surely important to validate the reproducibility of the technique in different samples from the same patient. However, since both disease progression and systemic treatment may impair the yield of the technique (e.g. by decreasing/increasing cfDNA or by selective pressure on mutated clones that may increase beyond the limit of detection during the treatment) the only possible way to obtain a “biological duplicate” in this setting is by analysis of different plasma samples acquired at very close time points before starting a new systemic therapy. We performed this type of repeated analysis on the first 10 patients in order to “calibrate” our results. Moreover, each time the result was discordant compared to tissue-based analysis or to previous plasma-based analyses, a confirmatory plasma sample was analysed.

This point was added to the M&M section.

5) In the introduction, the authors state that “using the Idylla Biocartis platform … is feasible in clinical practice.” One criteria to support this statement is the time for performing the analysis and interpret the data. How long does it take to prepare the samples, run the qPCR, analyze the data? How easy/difficult is it to interpret the data? How long does it take to train a person to perform the testing of RAS/BRAF mutational status? Please add the requested information to the discussion of the manuscript.

We wish to thank the reviewer for underlining another important point to the clinical feasibility of this methodology.

As a matter of fact, the only skill required for running the analysis once the plasma has been obtained by centrifugation is transferring 1 ml of plasma from the tube to each cartridge (one for KRAS and the other for NRAS/BRAF). Once the plasma is in the cartridge, the test can be started in the automated platform and only takes about 2 hours (85-130 min) to be performed. The results are readily available at the end of the run in the form of an automated report that includes: validity of the run (in case no DNA is detected, the test is considered invalid), presence or absence of mutation(s), type of mutation detected, Cq of the total KRAS/NRAS/BRAF, Cq of the specific mutation(s) detected.

For this reason, no specific training is required to either perform the test or interpret the data.

These information have been added to the manuscript.

6) Figure 2A: Sensitivity, specificity, and ppv can be used to describe the quality of a given marker. However, it is difficult to “classify” a single patient based on the analysis on the RAS/BRAF status with respect to his/her prognosis or response to therapy. In this context, the CMF value could be helpful. Does the “basal liver positive” (Table 1B) cohort contain only patients with liver metastasis only or does it contain patients with liver metastases only and patient with metastases in liver and other organs? a. Please define the “basal liver positive” cohort in Table 1B and add this information to the table legend. b. Please define the “basal liver positive” cohort in Figure 2A and add this information to the figure legend. c. Please use the same labels in Table 1B and Figure 2A and add the number of patients in each group to the figure legend of Figure 2A. d. If the “basal liver positive” cohort contains patients with liver metastases only and patient with metastases in liver and other organs, please change Figure 2A and distinguish between “liver positive” and “liver & other” by adding a third box to the diagram.

The “basal liver positive” cohort results from the sum of the basal patients with liver only disease (“liver only”) and those with metastases to liver and other organs (“liver&other”).

This issue was fixed; This issue was fixed; We corrected the labels between the table and the figure legend; In this particular analysis, we confronted patients without liver metastases (5 patients) with patients with liver metastases (9 patients: 2 with liver-only disease and 7 with liver & other organs metastases). Since the subgroup of patients with liver only disease consists of just 2 patients, no statistical analysis can be performed. This is the reason why we decided to include all the patients with liver metastases, collectively defined as “liver positive”.

Minor points: 1) Line 133: Please change “Figure 2 B” to “Figure 2B”. 

This typing mistake was corrected.

Reviewer 3 Report

This is a retrospective study reporting on a single institution experience of the analysis of circulating tumor DNA (ctDNA) using the IdyllaTM Biocartis test (liquid biopsy) to identify RAS/BRAF mutations in a series of 72 patients with metastatic colorectal cancer. The results of liquid biopsy was compared to that of tumor tissue NGS analysis.

General comments

Some important studies on the value of ctDNA and its concordance with tumor DNA to identify RAS/BRAF mutations in metastatic colorectal cancer are not cited in the present article. They should be mentioned and discussed as they reported different concordance rates and they give some explanations on the factors affecting concordance (Thierry AR Nature 2014; Thierry AR Ann Oncol 2017, Bachet JB Ann Oncol 2018). The authors found the higher concordance in treatment-naïve patients with liver metastases. This must be put into perspective with the results of the RASANC study that has already reported similar results (Bachet JB Ann Oncol 2018).

Specific comments

Chapter 2.4:

- there is one case of double positivity for codon 12 and codon 61 mutations. Comment on that phenomenon of multiple coexisting RAS mutations and discuss it against the results reported by Thierry AR et al. in Ann Oncol 2017.

- It would be interesting to give the response rate and duration of response of the 15 patients who were rechallenged with an anti-EGFR according to the occurence of KRAS mutation during the course of treatment.

Chapter 2.2

This chapter is completely redundant with the data of Table 1. Chose one or the other but not the two ways to report these results.

Learning from the discordant cases

- "these three patients did not have liver metastases, a condition that is more often associated with less ctDNA shedding in the blood...": refer to the RASANC study (Bachet Ann Oncol 2018) and comment on the problem of false negative cases of ctDNA in clinical practice regarding anti-EGFR treatment. Have the 4 FN cases be treated by anti-EGFR and if so what was the tumor response?

Chapter 3

What do you expect with CMF in addition to ctDNA in clinical practice? In which setting do you think it could have a clinical impact?

Chapter 4

- Concerning analyses on tissue, do the samples come from the primary colorectal tumor or from metastases (what site?)

Author Response

Dear Reviewer, 

thank you for considering our work for publication. 

Here are listed the responses to the comments as a point by point list, with the Authors' responses marked in Bold Italics

This is a retrospective study reporting on a single institution experience of the analysis of circulating tumor DNA (ctDNA) using the IdyllaTM Biocartis test (liquid biopsy) to identify RAS/BRAF mutations in a series of 72 patients with metastatic colorectal cancer. The results of liquid biopsy was compared to that of tumor tissue NGS analysis.

 General comments

Some important studies on the value of ctDNA and its concordance with tumor DNA to identify RAS/BRAF mutations in metastatic colorectal cancer are not cited in the present article. They should be mentioned and discussed as they reported different concordance rates and they give some explanations on the factors affecting concordance (Thierry AR Nature 2014; Thierry AR Ann Oncol 2017, Bachet JB Ann Oncol 2018). The authors found the higher concordance in treatment-naïve patients with liver metastases. This must be put into perspective with the results of the RASANC study that has already reported similar results (Bachet JB Ann Oncol 2018).

 Specific comments

Chapter 2.4:
- there is one case of double positivity for codon 12 and codon 61 mutations. Comment on that phenomenon of multiple coexisting RAS mutations and discuss it against the results reported by Thierry AR et al. in Ann Oncol 2017.

The Authors thank the Reviewer for the interesting comments on the manuscript.

We found a case of double positivity (KRAS codon 12 and 61). In the work by Thierry et al (Ann Onc, 2017) a higher percentage of samples harboured more than 1 mutation, but this reflects the higher sensitivity of IntPlexVR used for ctDNA analysis. In fact, tumor heterogeneity could lead to the development of many new mutations, not only in RAS and BRAF, but the clinical relevance of them is still debated. In the cited work, the higher is the threshold, the lower is the percentage of more-than-one mutation detected. Idylla has a lower sensitivity but this could probably eliminate a “background noise” of clinically irrelevant mutations. Our explanation to such a low prevalence of multiple mutations is that our unique patient with double mutation probably had two major clones with a distinct biological development.

A proof for this explanation derives from the curves of the qPCR analyses, that are available to the Authors using the Idylla explore tool. Two more cases of multiple mutations in KRAS gene are evident in the curves, but are not included in the final analyses because they do not reach the threshold of significance required by the automated report, and are thus deemed as aspecific.

These observations have been included in the manuscript.

- It would be interesting to give the response rate and duration of response of the 15 patients who were rechallenged with an anti-EGFR according to the occurence of KRAS mutation during the course of treatment.

Concerning the response rate in patients who rechallenged with an anti-EGFR, data on ORR and DoR are still immature and will be shown in a future work by our research group.

 Chapter 2.2

This chapter is completely redundant with the data of Table 1. Chose one or the other but not the two ways to report these results.

We think it is important to stress these results in the main text, whilst the table is for rapid and schematic consultation.

Learning from the discordant cases

- "these three patients did not have liver metastases, a condition that is more often associated with less ctDNA shedding in the blood...": refer to the RASANC study (Bachet Ann Oncol 2018) and comment on the problem of false negative cases of ctDNA in clinical practice regarding anti-EGFR treatment. Have the 4 FN cases be treated by anti-EGFR and if so what was the tumor response?

False negative could be a big problem using Idylla, but mainly in patients with low tumor burden, thus limiting the reliability of the results in this setting. In fact, as confirmed by multiple authors, patients without liver metastases have lower ctDNA than patients with liver metastases and thus mutations could be missed by Idylla for its low sensitivity. These observations and the relative reference have been included in the manuscript

The 4 FN did not and could not receive anti-EGFR therapy since  the presence of KRAS or NRAS mutations on standard-of-care tissue analysis on FFPE specimen does not allow the use of anti-EGFR agents in clinical practice, in Italy. Moreover, liquid biopsy data cannot be used yet to guide anti-EGFR therapy i outside a clinical trial.

 Chapter 3

What do you expect with CMF in addition to ctDNA in clinical practice? In which setting do you think it could have a clinical impact?

The automated report by the Idylla platform does not include a dose of the ctDNA, which is thus not available for clinical use. For the purpose of our study, we propose the Circulating Mutational Fraction (CMF) as a surrogate biomarker for the clonality of a specific RAS/BRAF mutation  in a patient (i.e. mutational allele fraction),  since it requires a mutation to be calculated. Moreover, the relative abundance of non-tumor cfDNA can impair the use of CMF as a surrogate for ctDNA, in particular when comparisons are performed between different patients.
For these reasons, as we propose in the manuscript, the CMF is better used as a dynamic marker of the emergence of RAS-mutated clones and their clonal evolution for patients who receive anti-EGFR therapy. In fact, it is known that the detection of RAS mutated clones can anticipate clinical progression by several months (up to 10 in Siravegna G et al, Nat Med 2015, ref.25). In this frame, serial testing for each patient undergoing anti-EGFR therapy could help defining the onset of resistance and evaluate new therapeutic approaches to the disease.
The manuscript was edited in order to better underline these points.

 Chapter 4
- Concerning analyses on tissue, do the samples come from the primary colorectal tumor or from metastases (what site?)

This is an interesting point. In our clinical practice, we usually analyze the primary tumor (biopsy or surgical specimen), because of the high molecular concordance between primary CRC tumour and metastases. For only 2 patients in our series, analysis had been performed on metastases because primary tissue was not available for the analysis, and we used lung and liver metastases samples, respectively.

Reviewer 4 Report

The authors analyzed a cohort of 72 metastatic colorectal cancer patients evaluated with liquid biopsies and standard tissue-based next generation sequencing (NGS). There were two groups well differentiated. One group constitute 42 untreated patients (26 RAS or BRAF mutant patients and 17 double WT) and 30 pre-treated patients (15 pre-treated with and 15 pre-treated without anti-EGFR drugs). Major contributions are a high concordance between NGS (81.9% 59/72) on FFPE and liquid biopsies (Idylla-Biocartis). In addition the authors analyzed circulating mutational fraction (CMF) as a composed biomarker (mutated DNA over the internal control calculated as 2-Cqx100) and correlated CMF with the abundance of total ctDNA and the mutational allele fraction /MAF).

Although the study has some novelty such as CMF analysis, there is between NGS (81.9% 59/72) on FFPE some major weakness that should be clarified.

First, the analysis of concordance should be done separately between the two cohorts of patients (42 naive and 30 pre-treated). All the analysis between FFPE and liquid biopsies including CEA, liver metastases but also other bulky parameters such as number of organs and LDH levels should be done separately in both cohorts. The status of NGS on FFPE in 30 pre-treated patients is not shown. These cases are all NGS on FFPE 2WT and therefore there are 10/30 conversions?. In pre-treated liquid biopsies positive samples 5/15 positive were treated with anti-EGFR compounds and 5/15 was positive, treated with only chemo. Is it true?. Please clarify. I suggest to put in Table 1 TP, FP, FN, concordance, sensitivity, specificity and PPV in both cohorts with previous treatment (and types) and patients characteristics (including ECOG PS, LDH, number of metastatic organs, CEA, alkaline phosphatase and organ location). Consider also to eliminate Table 1B.

Second, CMF at least with Idylla test it measures only the abundance of total ctDNA but not discriminate the mutation allele fraction. The authors can speculate if mutation allele fraction was obtained with FFPE determination in synchronic disease if the sum of MAF plus tumor mutational burden correlated with liquid biopsy sensitivity, but authors did not make this analysis. These issues should be re-written in the discussion.

Round 2

Reviewer 2 Report

The authors have addressed all points mentioned in the first round of the reviewing process to my satisfaction. In particular, I am pleased by the additional information provided by the authors regarding the Materials & Methods section as well as the clear definition of the groups presented in Table 1 and Figure 2. For the final version of the manuscript, please include information provided in your answer to major point 1 in the Materials & Methods section: “…with a maximum of 9 ml; The mean yield of plasma was about 3 ml for each patient.” Once this minor point is addressed, I recommend to publish the manuscript.

Reviewer 4 Report

No comments